# Micro RNA Transcriptome Profile in Canine Oral Melanoma

**DOI:** 10.3390/ijms20194832

**Published:** 2019-09-28

**Authors:** Md. Mahfuzur Rahman, Yu-Chang Lai, Al Asmaul Husna, Hui-wen Chen, Yuiko Tanaka, Hiroaki Kawaguchi, Noriaki Miyoshi, Takayuki Nakagawa, Ryuji Fukushima, Naoki Miura

**Affiliations:** 1Veterinary Teaching Hospital, Joint Faculty of Veterinary Medicine, Kagoshima University, Kagoshima, Kagoshima 890-0065, Japan; sajib.mahfuz.bau@gmail.com (M.M.R.); cyliang001@gmail.com (Y.-C.L.); asmaul.hausna@gmail.com (A.A.H.); 2The United Graduate School of Veterinary Science, Yamaguchi University, Yamaguchi, Yamaguchi 753-8515, Japan; 3Joint Graduate School of Veterinary Medicine, Kagoshima University, Kagoshima, Kagoshima 890-0065, Japan; helenshe2001@yahoo.com.tw; 4Laboratory of Veterinary Surgery, Graduate School of Agricultural and Life Sciences, The University of Tokyo, Bunkyo City, Tokyo 113-8654, Japan; yui.tanaka.810@gmail.com (Y.T.); anakaga@mail.ecc.u-tokyo.ac.jp (T.N.); 5Hygiene and Health Promotion Medicine, Kagoshima University Graduate School of Medicine and Dental Science, Kagoshima, Kagoshima 890-8544, Japan; k3038952@kadai.jp; 6Department of Veterinary Histopathology, Joint Faculty of Veterinary Medicine, Kagoshima University, Kagoshima, Kagoshima 890-0065, Japan; miyoshi@vet.kagoshima-u.ac.jp; 7Animal Medical Centre, Tokyo University of Agriculture and Technology, Tokyo, Tokyo 183-8538, Japan; ryu-ji@cc.tuat.ac.jp

**Keywords:** microRNAs, next-generation sequencing, dog, melanoma

## Abstract

MicroRNAs (miRNAs) dysregulation contribute the cancer pathogenesis. However, the miRNA profile of canine oral melanoma (COM), one of the frequent malignant melanoma in dogs is still unrevealed. The aim of this study is to reveal the miRNA profile in canine oral melanoma. MiRNAs profile of oral tissues from normal healthy dogs and COM patients were compared by next-generation sequencing. Along with tumour suppressor miRNAs, we report 30 oncogenic miRNAs in COM. The expressions of miRNAs were further confirmed by quantitative real-time PCR (qPCR). Pathway analysis showed that deregulated miRNAs impact on cancer and signalling pathways. Three oncogenic miRNAs targets (miR-450b, 301a, and 223) from human study also were down-regulated in COM and had a significant negative correlation with their respective miRNA. Furthermore, we found that miR-450b expression is higher in metastatic cells and regulated *MMP9* expression through a PAX9-BMP4-MMP9 axis. In silico analysis indicated that miR-126, miR-20b, and miR-106a regulated the highest numbers of differentially expressed transcription factors with respect to human melanoma. Chromosomal enrichment analysis revealed the X chromosome was enriched with oncogenic miRNAs. We comprehensively analyzed the miRNA’s profile in COM which will be a useful resource for developing therapeutic interventions in both species.

## 1. Introduction

One person dies every hour from melanomas, new melanoma cases have increased by 53% in the US [1], and the WHO reported that 132,000 new melanoma cases are diagnosed every year in the world. These reports clearly confirm the importance of melanoma studies. Molecular studies have enriched the definition of melanoma sub-types [2,3]. The triple wild-type (TWT) subtype bears the features that underlie non-cutaneous melanoma, including mucosal melanoma [3,4]. Human mucosal melanoma is more aggressive with less favourable prognosis than other subtypes.

Previous studies suggested dog melanoma as a natural model for human melanoma [5,6]. Malignant melanoma is frequent in dog and the majorities are in the oral mucosa. Oral melanoma in dogs is considered a typical model for non-UV or TWT melanoma in human [2,3]. Dog melanoma genes have the same mutations or aberrant expression as human melanoma genes, *BRAF*^V600E^, *NRAS* (Q61) [7], *PTEN* [5], and *KIT* [8]. Besides the protein-coding RNAs, non-coding RNAs (ncRNAs) also have important roles in gene regulation. Among them, small non-coding RNAs have widespread regulatory functions in human diseases, and microRNAs (miRNAs) are now in phase I trials to treat human hepatitis, diabetes, lymphoma, scleroderma, and lung cancer [9].

Next-generation sequencing (NGS) has been used widely to study miRNA, including their role in human melanoma. For dogs to be a useful therapeutic preclinical model, knowing the miRNA profile in dog melanoma is important. There are few reports of tumour suppressor miRNAs in canine oral melanoma (COM) [10,11], and studies of miRNA profiles of human TWT or mucosal melanoma are scarce. Moreover, the global deregulated miRNAs expression profile of COM is still unrevealed. So we aimed to use next-generation sequencing to study global aberration of miRNAs in COM bearing following three objectives: (1) Comprehensively profile miRNA expression pattern in COM, (2) Validation of findings in the COM tissue samples by using qPCR, and (3) Investigate the gene-regulatory function and molecular pathways of differentially expressed miRNAs. In our study, we obtained the miRNA profile of eight COM tissue samples by NGS which were further validated by qPCR. We found several miRNAs were differentially expressed. We also explored a new function of miR-450b. The impact of global changes in the miRNA profile was shown by Kyoto Encyclopedia of Genes and Genomes (KEGG) pathway analysis. Finally, a common miRNA and transcription factor (TF) network were constructed and analyzed to find the most important miRNAs for the regulation of TFs expression between dog and human melanoma.

## 2. Results

### 2.1. Small RNA Profile of Canine Oral Melanoma

To investigate the miRNA profile in COM, RNA from three normal oral tissue samples from healthy dogs (hereafter referred to as ‘‘normal’’) and eight samples from dogs with COM was sequenced (Appendix A). After adapter trimming and quality check, we obtained 51 and 142 million clean reads from normal and melanoma libraries, respectively (Appendix A). Sequences were submitted to the SRA database (PRJNA516252). Length distribution analysis showed 90% and 82% of clean reads in the normal and melanoma libraries, respectively, were 20–24-nt long, indicating an alteration in the small RNA profile (Appendix A). We annotated the reads using miRBase or Ensembl dog and human ncRNAs (see methods), witch 92.5% and 84.23% of the reads in normal and melanoma, respectively, were annotated (Appendix A). Among the annotated reads miRNAs were the most abundant small RNAs. Interestingly, the percentage of other ncRNAs reads (Mt-rRNA, Mt-tRNA, linc-RNA, sno-RNA, snRNA, misc-RNA, rRNA, other miR) was two times more in the melanoma group (Figure 1a). SnRNA, snoRNA, and mitochondrial tRNA-derived small RNA fragments were the most altered between the two groups. As miRNAs were the most abundant we analyzed the miRNAs further.

We found significant differences in the chromosome distribution of the annotated miRNAs mapped reads between the normal and melanoma groups (Figure 1b), implying altered global miRNA profiles in the melanoma group. We annotated 542 miRNAs in both groups, among them 64 miRNAs were differentially expressed (Figure 1c). The top 20 highly expressed miRNAs made up >70% of the total reads that were annotated to miRBase (Figure 1d). Among them, 12 miRNAs were common between the groups. We selected the rank of the miRNAs on the basis of their expression. The rank orders of miRNA’s were different in melanoma than normal (Figure 1d). Four of the top 10 highly expressed miRNAs in melanoma were not in the top 10 of the normal group. Importantly, miR-21, which is a well-characterized miRNA oncogene frequently found to be over-expressed in various malignancies, was ranked one in melanoma and 12 in normal. Also, miR-22, miR-148a, and miR-186, all of which have been reported to be oncogenic, but not statistically significant in our study also changed their rank within top 10 in melanoma [12,13,14]. However, the ranks of some known anti-oncogenic miRNAs were much lower in melanoma than in normal. For example, miR-203 and miR-205, which were reported to be anti-oncogenic in melanoma [10] were ranked 6 and 7 in normal, were outside the top 20 in melanoma. So, melanoma reduced expression of anti-oncogenic miRNAs while taking favour of others highly expressed miRNAs by remodelling their expression position according to their target functionality.

### 2.2. Global miRNAs Expression in Canine Oral Melanoma

Unsupervised hierarchical clustering and principal component analysis were performed for all the differentially expressed miRNAs (FC > 1, without considering the FDR and mean read count) to evaluate similarities between the studied samples at the global level (Figure 2a, and Appendix A). After applying stringent filtering criteria (Fold change >2, FDR <0.05, and miRNA mean read counts in either normal or melanoma >50), we obtained 30 up- and 34 down-regulated miRNAs (Figure 2b, Appendix A). The heatmap and clustering tree revealed a distinct miRNA expression pattern between the groups. The principal component analysis and clustering tree showed that the differentially expressed miRNA were enough to distinguish the two groups, and heatmap showed the miRNA expression patterns were similar within a group. Data showed there were significant changes in the miRNA profile in COM.

### 2.3. Validation of miRNA Expression

We selected 20 differentially expressed miRNAs for validation by qPCR by 12 normal oral and 17 melanoma tissue samples (Figure 2c,d and Appendix A). We selected these 20 miRNAs on the basis of three different criteria: (1) miRNAs those were not reported or validated previously (miR-450b, 301a, 140, 542, 223, 190, 429, 96, 375,183, 200b, 141), (2) miRNAs those were reported or validated previously (miR-21, miR-122, 383 and 143) [15], (3) miRNAs those were beyond (numerically close) to the stringent filtering criteria (miR-122, miR-34a, miR-338, miR-182, and miR-1). Among the up-regulated miRNAs, miR-450b, miR-223, miR-140, miR-542, and miR-383 showed >10-fold change and miR-301a, miR-21, and miR-190 showed >5 fold-change (Figure 2c and Appendix A). Among the down-regulated miRNAs, miR-429, miR-200b, miR-141, and miR-375 showed <−10-fold change, miR-96 showed <−3 fold-change, and miR-183 and miR-143 showed <−2-fold change (Figure 2d and Appendix A). There was significant positive correlation of fold differences between the NGS and qPCR results (Figure 2e), and the expression levels of miR-122, miR-34a, miR-338, miR-182, and miR-1 (Figure 2c,d and Appendix A) which were beyond our stringent filtering criteria, showed similar trends between NGS and qPCR indicates the strength of our filtering criteria. These results confirm the expression of several oncogenic and tumour suppressor miRNAs in COM revealed by NGS and validated by qPCR.

### 2.4. Gene Regulatory Function of Oncogenic miRNAs

MiRNAs do their function by negatively regulating the gene expression in the mRNA and protein level. To know the differentially expressed miRNAs function, we need to focus on the expression of their respective target genes. However, it is unfeasible to explore all the differentially expressed miRNAs targets in a single study. So, we selected three miRNAs (miR-450b, miR-301a, and miR-223) to know their gene regulatory function because to our knowledge miR-450b has no report in melanoma and other two miRNAs are less studied in human melanoma and no report in COM. We selected *PAX9*, *NDRG2*, and *ACVR2A* as targets of miR-450b, miR-301a, and miR-223, respectively, from previous studies where they validated miRNA-mRNA binding experimentally [16,17,18]. The binding sites in the 3′ un-translated regions of these genes predicted by TargetScan 7.2 were conserved between human and dog (Appendix A). So, we hypothesized that similar phenomenon (miRNA-mRNA binding) may exist in the canine oral melanoma. As in our experiment miR-450b, 301a and 223 was up-regulated so if our hypothesis is true the target genes must be down-regulated and they should have a strong negative correlation with the respective pairs. Our qPCR results showed significant down-regulation of *PAX9*, *NDRG2*, and *ACVR2a* in melanoma compared with normal, and the relative expression of the respective miRNA–mRNA pairs showed significant negative correlation (Figure 3a,b). This inverse relationship indicates the miRNAs may bind the respective mRNA targets like previous human studies and suppress their expression in melanoma. In addition, study reported that PAX9 down-regulation decrease BMP4 expression which can increase *MMP9* expression [19,20]. So we investigated the PAX9-BMP4-MMP9 axis in our study. Our qPCR results showed that *BMP4* was down-regulated and *MMP9* was up-regulated in melanoma tissue samples (Figure 3c). MMP9 is required for the degradation of the extracellular matrix, which is a prerequisite for tumour invasion and positively correlates with tumour metastasis. So we expected high MMP9 expression in metastatic cells along with miR-450b. Therefore, we further investigated the relative expressions of miR-450b, *PAX9*, *BMP4*, and *MMP9* in two COM cell lines: KMEC established from primary oral melanoma and LMEC from metastatic mandibular lymph node of oral melanoma [21]. qPCR analysis showed that miR-450b was up-regulated and *PAX9* and *BMP4* were significantly down-regulated in LMEC compared with KMEC. Conversely, as expected, *MMP9* was significantly up-regulated in LMEC compared with KMEC, as shown in Figure 3d. So, we concluded that up-regulation of miR-450b and down-regulation of *PAX9* and *BMP4* were interconnected with the high *MMP9* expression in metastatic melanoma cells (Figure 3e).

### 2.5. Gene Ontology and KEGG Pathway Analysis of the Differentially Expressed miRNAs

To determine the global function of the differentially expressed miRNAs, we predicted their target genes by overlaying the results obtained using TargetScan and miRDB. We detected 2555 and 2464 target genes of the down- and up-regulated miRNAs, respectively (Appendix A). We functionally annotated the target genes by assigning gene ontology (GO) terms and KEGG pathways. Target genes of the down-regulated miRNAs were analysed against cancer and other databases (Appendix A). We found oncogenic genes related terms were enriched which indicates these genes have an oncogenic function. From GO analysis we found protein modification (e.g., phosphorylation, transcription, or repression from DNA), extracellular matrix, and receptor signalling GO terms were assigned for the target genes of down-regulated miRNAs. It indicates down-regulated miRNAs inhibit their target genes to maintain target genes terms related function in normal condition which was disrupted in melanoma due to the miRNA down-regulation. Target genes of the up-regulated miRNAs were involved mainly in the protein ubiquitination system because ubiquitin-dependent protein catabolic process and ubiquitin-protein ligase activity terms were enriched in GO (Table 1). Ubiquitylation is long known for driving cell cycle transition and ubiquitin ligase has relation to the cell cycle control. This indicates that up-regulated miRNAs may be involved in cell cycle control by ubiquitination system.

The KEGG pathway analysis of the target genes revealed that the down-regulated miRNAs were involved in tuning of several signalling pathways that are known to be disrupted in diseases, and the up-regulated miRNAs were related to remodelling of extracellular matrix organization, packaging, circadian entrainment, recycling and modification of receptors, proteins, chemokines and enzymes in favour of disease progression (Table 2).

### 2.6. miRNA–Transcription Factor Interaction Network between Dog and Human Melanoma

Previous human melanoma studies indicate that several differentially expressed miRNAs have similar trend of expression with COM [22,23,24,25,26]. Transcription factors like MITF can play a critical role in melanoma [27,28]. So, we were interested to find a common miRNA–TFs co-regulatory network in human and dog for melanoma. Moreover, miRNAs those can target maximum number of TFs will be considered to be important for melanoma-related TFs regulation. We considered the same seed sequences miRNAs between human and dog from our study and the miRNAs target (genes) orthologues TFs that were differentially expressed in human melanoma (GSE31909) were selected for network construction. A total of 34 up-regulated and 33 down-regulated TFs were obtained from GSE31909 (Appendix A). We constructed two networks, one using down-regulated miRNAs and up-regulated TFs, and the other using up-regulated miRNAs and down-regulated TFs (Figure 4a,b). See methods for details. We measured the degree and betweenness centrality of the networks to detect the key miRNAs that can regulate maximum TFs in both groups. Nodes that had higher centrality values than average were considered to influence the network biologically.

In the down-regulated miRNA–TF regulatory network (Figure 4a), the miR-126, miR-183, and miR-200 families, let-7 family members had higher degree centralities than average. Among them, miR-126 had the highest centrality (Appendix A).

In the up-regulated miRNA–TF regulatory network (Figure 4b), miR-130 family, and miR-9, miR-20b, miR-371, miR-106a, miR-450b, miR-21, and miR-424, had higher degree centrality than average. Among the miRNAs, miR-20b and miR-106a had the highest centralities (Appendix A). So it indicates that among the differentially expressed miRNAs, miR-126, miR-20b and miR-106a are the most potent to regulate the maximum number of TFs in melanoma.

### 2.7. Differential miRNA Chromosomal Enrichments

Studies reported that more than 50% of miRNA genes are encoded in the cancer-associated regions or fragile sites or chromosomal breakpoints which are frequently absent, amplified or rearranged in tumour specimens[29,30]. Thereby, the dysregulation of miRNA expression occurred in tumour. For example, miR-15a/16-1 is located in genomic locus containing tumour suppressor that is frequently deleted in leukaemia [31]. So to investigate the melanoma-associated regions or fragile sites it’s important to understand in which chromosome the differential miRNAs reside. We analyzed the chromosomal locations of all 542 annotated miRNAs from miRBase. Stem-loop or mature sequences were mapped against the dog genome to obtain locations for the hsa-miRs (miRNAs that were annotated by the human sequence). Among the 542 miRNAs, 70 (12.84%) were in the X chromosome, and 14 (2.57%) and 24 (4.40%) were in chromosomes 4 and 5, respectively. Three chromosomes were focused because most of differentially expressed miRNAs were encoded in these chromosomes (Figure 5a,b). Out of 30 up-regulated miRNAs, 11 (34.4%) were in X, an about 2.8-fold significant enrichment (*p* = 5.6 × 10^−4^). Among the down-regulated miRNAs, four in chromosome 4 (*p* = 0.008, enrichment = 4.55-fold) and six in chromosome 5 (*p* = 0.001, enrichment = 4.55-fold) were enriched.

Among the 11 up-regulated miRNAs in Chromosome X, nine miRNAs encode from two clusters: mir-106a/18b/20b/19b-2/92a-2/363 and mir-424/503/542/450a-2/450a-1/450b. Other members in the clusters (miR-19b-2, miR-92a-2, and miR-503) are not listed among the up-regulated miRNA list because they did not meet our stringent criteria, but the changes in their expression were similar to other members of the clusters, except miR-92a-2. Two miRNAs, miR-223 and miR-421, are encoded separately as single genes.

Among the four miRNAs in Chromosome 4 two from miR-143/145 cluster, and rest miR-1271 and miR-1260a encode as a single gene. Rest six down-regulated miRNAs from chromosome 5 encode from two clusters, the miR-200 family and miR-99a-2/let-7a-2, and miR-101 as a single gene. All the family miRNAs had similar expression patterns.

These results are consistent with that study, found clustered miRNAs stay and act together [32]. It also indicates that in chromosome X, 4 and 5 most of the differentially expressed miRNAs were from clusters and their other cluster members also have a similar trend of expression. This may recognize COM specific chromosomal fragile sites.

## 3. Discussion

Despite dogs being considered as a preclinical model for human melanoma [6], until now, the global miRNA profile was not fully revealed. In this study, we comprehensively analyzed the miRNA profile in COM. The expression levels of miRNAs studied previously [10,11] and our recently reported oncogenic miRNA [15] expression were consistent with those of the present study. Moreover, we detected several differentially expressed miRNAs that have not been reported previously (Appendix A).

Some of the differentially expressed miRNAs (up-regulated miR-301a, 130, 383, 21, 454, 335, 132, 423,424, 146b, 9, 20b and down-regulated let-7a, 7b miR-126, 125a, 183, 26b, 29c, 152, 31, 145, 141, 205, 203, 200, 101) were reported in human melanoma [22,23,24,25,26]. The expression trends of these miRNAs correlated well between human melanoma and COM. This indicates an overlap of miRNomes between the species and can be used as a model for human miRNA therapeutics development. It also affirms that dogs share much of their ancestral DNA with humans [33]. To further understand the functions of the miRNAs compared to humans, the targets of miR-450b (*PAX9*), miR-301a (*NDRG2*), and miR-223 (*ACVR2A*) which were reported previously [16,17,18] in human were analyzed. We found, *PAX9* and *NDRG2*, which were down-regulated in human and mouse melanoma, also were down-regulated in COM [34,35]. The expression of the miR-450b–*PAX9* and miR-301a–*NDRG2* pairs was significantly negatively correlated, which supports two studies that reported miR-450b and miR-301a can bind and suppress PAX9 and NDRG2 activity, respectively [16,17]. *ACVR2A* is reported here for the first time in melanoma with significant negative correlation with miR-223. These results suggest that the tumour suppressive function of PAX9, NDRG2, and ACVR2a were disrupted by miR-450b, miR-301a, and miR-223, respectively, to maintain oncogenic characteristics in COM as like the human studies. Moreover, as previous study reported that PAX9 is down-regulated and miR-301a, 223 is up-regulated in human melanoma [22,35,36]. It also indicates that there are similarities between human and canine melanoma in respect of PAX9, miR-301a and 223 expressions.

The predicted binding site of miR-450b-PAX9 is conserved between human and dog (Appendix A). BMP4, a downstream of PAX9, was suppressed when miR-450b degraded the function of PAX9, resulting in an increase in MMP9. Previous studies also showed that suppression of PAX9 decreased BMP4 expression and subsequently increased MMP9 [19,37]. Our study revealed that, in COM, this axis is also maintained and interconnected with high expression of miR-450b. Additionally, high MMP9 expression in metastatic melanoma cells may be maintained by this axis.

Axon guidance, endocytosis, and regulation of actin cytoskeleton and pathways in cancer are common pathways between human and dog melanoma regulated by miRNAs [26]. With canine cutaneous melanoma, only PI3K-Akt signalling, focal adhesion, and ECM-receptor interaction pathways are common [38]. This is not surprising because there are molecular differences between cutaneous and mucosal melanomas, so different pathways are likely to be affected [2,3].

TFs can regulate single or multiple gene expressions, so investigation of miRNAs that influence TFs could be more meaningful. MiR-126 has maximum influence over eight TFs that were up-regulated in melanoma. Although, low miR-126 expression was found to have poor prognostic value in several cancers [39], up-regulated miR-20b and miR-106a influenced 11 and 10 TFs, respectively. The miR-20b seed sequence is similar to that of human miR-17-5p, and miR-106a belongs to the miR-17-92 family. Over-expression of hsa-miR-17 and miR-106a is a good predictor of poor overall survival in several human cancers [40], indicating these miRNAs may be a prognostic marker and also a good therapeutic option in both species.

Chromosomal enrichment showed that the X chromosome harboured up-regulated miRNAs. In human melanoma, X-linked miRNAs are also enriched. Women have consistent advantageous prognosis in melanoma compared with men [41]. However, in mucosal melanoma, the incidence is higher in female [42]. Also, breast cancer has X chromosome-linked differential miRNA enriched in women [43]. To our knowledge, until now, the correlation between X-linked miRNA and poor survival has not been explained. However, in humans, a high number of miRNAs related to cancer located in X chromosome compared to Y [44]. Also study shows 52.5% of human miRNA genes are encoded in the cancer-associated regions or fragile sites or chromosomal breakpoints [30]. As a result miRNAs are frequently absent, amplified or rearranged in tumour specimens [30,31]. So, the speculation that miRNA clusters or family members are co-regulated with melanoma-related genes to achieve a regulatory net outcome on a cell or environment is a reasonable explanation of the enrichment of X chromosome-linked differentially expressed miRNAs. On the other hand, down-regulated miRNAs enriched in chromosome 4 and 5. Previous studies showed that miR-15a/16-1 is down-regulated in leukaemia, due to the deletion of genomic locus containing a putative tumour suppressor-containing region, [31]. Also let-7g/mir-135-1 deletion are reported in varies human malignancies [30]. So, further study regarding the down-regulation of two cluster miR-143/145 and 200 families may be interesting to find putative region in the respective chromosomes.

One drawback in our experiment might be the use of less normal samples in NGS screening. However, we overcome the issue by the qPCR validation of 20 differentially expressed miRNAs within 12 normal and 17 melanoma samples. Moreover, all the normal samples were from healthy laboratory beagle dogs which minimize the limitation of diversity regarding normal samples.

Our study comprehensively established a miRNA profile of COM that has not been previously implicated. We have also shown the significance of miR-450b over-expression in melanoma metastatic cells and future studies are necessary to evaluate the others. Furthermore, we are able to report some melanoma-related miRNAs that are also important in human. Besides, as dog oral melanoma is considered as a typical model for non-UV or TWT melanoma in human our findings will give an insight into the miRNA expression of TWT and mucosal melanoma.

## 4. Materials and Methods

### 4.1. Clinical Samples and Canine Melanoma Cell Lines

COM tissue specimens were acquired from tumours excised from dogs that had undergone surgery at the Veterinary Teaching Hospital of Kagoshima University. Informed consents were obtained from dog owners. COM patient’s information is presented in Appendix A. Normal oral tissues were collected from healthy laboratory beagle dogs (age range 8–10 years) at Kagoshima University. Experimental conditions and design were approved by Kagoshima University and Veterinary Teaching Hospital ethics committee (KV004; 18.04.2011). All experimental methods were carried out in accordance with the approved guidelines and regulation.

Tissue samples were collected immediately after excision from dogs that had undergone surgery. The diagnosis was confirmed histopathologically by the hospital. The tissue specimens were placed in RNA*later* (AM7021, Invitrogen, Carlsbad, CA, USA) immediately after isolation and stored at −80 °C after overnight incubation at 4 °C.

Dog melanoma cell lines KMEC and LMEC were stored in freezing medium (039-23511, CultureSure, Fujifilm Wako Pure Chemical Corporation, Osaka, Japan). Cell lines were cultured according to the procedure described previously [21]. Cells were grown until confluence and then RNA was extracted for evaluation.

### 4.2. RNA Extraction and Sequencing

A mirVana RNA Isolation kit (AM1560, Thermo Fisher Scientific, Waltham, MA, USA) was used for RNA isolation according to the Manufacture’s standard protocol. The total RNA concentration was measured using a NanoDrop 200c spectrophotometer (ND2000C, Thermo Fisher Scientific). The RNA quality and integrity were assessed with an Agilent 2100 Bioanalyzer (G2939BA, Agilent Technologies, Santa Clara, CA, USA). The RNA Integrity Number (RIN) mean value was 8.8 (range 7–10) for tissue samples and 9.9 (range 9.6–10) for the KMEC and LMEC cell lines.

Following RNA isolation and quality measurement, samples were sequenced by the Hokkaido System Sciences Company (Hokkaido, Japan). Briefly, small RNA (sRNA) libraries were constructed using 1 µg of total RNA with the TruSeq Small RNA Library Preparation kit (Illumina, San Diego, CA) following the manufacturer’s protocol. After obtaining the sRNAs (18–30 nt) from the total RNA, 5′ and 3′ adaptors were ligated to the sRNAs. Then, reverse transcription followed by amplification was performed to create cDNA constructs. A gel purification step was applied to purify the amplified cDNA constructs for cluster generation and Illumina/Hiseq2500 sequencing analysis by the Hokkaido System Science Co., Ltd. (Hokkaido, Japan). The high-quality cleaned reads that we obtained from the company are shown in Appendix A (Phred score > 34). The raw sequences have been submitted to NCBI sequence read archive (SRA) database under accession number PRJNA516252

### 4.3. Bioinformatics Analysis of Small RNA Reads

The RNA sequencing data were imported into the CLC Genomics Workbench (CLC Bio, Qiagen, Germany) as recommended in the manufacturer’s manual (http://resources.qiagenbioinformatics.com). Normalization of reads, quality, ambiguity, and adapter trimming as well as quality control was performed using the CLC Genomics Workbench (versions 9 and 10). Briefly, the sequencing generated about 103 and 266 million reads from the normal and melanoma libraries, respectively, with single-end reads (Appendix A). We performed a two-step trimming process to remove adapters and other contaminants. In step one, we aimed to remove low quality, ambiguous nucleotides, 3′ adapters, and short (>15 nt) and long reads (>29 nt). In step two, we removed contaminated sequences, 5′ adapters, and the Illumina stop oligo sequence (5′-GAATTCCACCACGTTCCCGTGG-3′). Finally, we obtained about 51 and 142 million reads from the normal and melanoma libraries, respectively, for further analysis of the small RNA reads (Appendix A). Clean reads were analyzed according to the small RNA analysis guideline of the CLC Genomics Workbench. Briefly, the CLC Genomics Workbench was used to extract and count the small RNA from the clean reads and then compare them to databases of miRNAs and other small RNA databases for annotation. Sequence/fragment counts were used as the expression values for the miRNAs/small RNAs in the libraries.

To annotate the small RNA other than miRNA, CLC bio uses two other reference databases (Canis_familiris.canfam3.1.ncrna and Homo_sapiens.GRCh37.ncrna) from ensemble to annotate sequences that had no matches in miRBase [45]. Differential expression between the two groups was followed the EDGE (empirical analysis of differential gene expression) analysis within the CLC bio.

### 4.4. Edge Analysis

EDGE follows the exact test developed by Robinson and Smyth for two-group comparisons [46]. The exact test counts data that follow a negative binomial distribution and compares the counts in one set of count samples against the counts in another set of count samples. The variability of each group also is taken into account. The original count data are used because the algorithm assumes that the counts on which it operates are negative binomially distributed. We used the default parameters throughout the analysis. Fold change was calculated from the estimated average count per million (cpm) from each group. The estimated average cpm is derived internally in the exact test of the algorithm. Fold change indicates the difference in average cpm values between the groups. The FDR is based on the p-value of the exact test.

### 4.5. Expression Analysis by qPCR

To measure the expressions of miRNAs and mRNAs by qPCR we used TaqMan microRNA and gene expression assays (Thermo Fisher Scientific). Total RNA (2 ng) was reverse transcribed to cDNA using a TaqMan MicroRNA Reverse Transcription kit (4366597, Thermo Fisher Scientific) according to the manufacturer’s protocol. qPCR was performed using a TaqMan First Advanced Master Mix kit and a one-step plus real-time PCR system (Thermo Fisher Scientific). Thermal cycling was performed according to the manufacturer’s instructions. All experiments were performed in duplicate. The Cq values of RNU6B in the normal and melanoma samples were consistent between the groups, so RNU6B was used as an internal control to calculate miRNA expression. ΔCq was calculated by subtracting the Cq values of RNU6B from the Cq value of the target miRNA. ΔΔCq was calculated by subtracting the mean target miRNA ΔCq value from the ΔCq value of the normal and melanoma samples. The expression level was evaluated using the 2^−∆∆Cq^ method [47]. qPCR reactions of undetermined Cq were assigned Cq = 36 cycle. TaqMan MicroRNA assays were used in this study can quantitate mature miRNAs. MiRNA primer IDs were as follows: RNU6B (ID: 001093), miR-450b (ID: 006407), miR-301a (ID: 000528), miR-140 (ID: 007661), miR-383 (ID: 000573), miR-542 (ID: 001284), miR-223 (ID: 000526), miR-190 (ID: 000489), miR-21 (ID: 000397), miR-122 (ID: 002245), miR-34a (ID: 000426), miR-338 (ID: 000548), miR-429 (ID: 001077), miR-96 (ID: 000186), miR-375 (ID: 000564), miR-183 (ID: 002269), miR-182 (ID: 002334), miR-1 (ID: 000385), miR-143 (ID: 002249), miR-200b (ID: 002251), and miR-141 (ID: 000463).

For the target gene mRNAs, 250 ng RNA was reverse transcribed to cDNA using ReverTra Ace qPCR RT master mix with gDNA Remover (FSQ-301, Toyobo, Osaka, Osaka Prefecture, Japan). The qPCR procedure was the same as that used for the miRNA experiments. The 2^−∆∆Cq^ method also was used to calculate the expression. *GAPDH* was used as an internal control. The TaqMan gene expression assay was used in the experiments. The gene IDs were as follows: *GAPDH* (ID: Cf04419463_gH), *PAX9* (ID: Cf02705737_m1), *MMP9* (ID: Cf02621845_m1), *BMP4* (ID: Cf01041266), *NDRG2* (ID: Cf02631635_m1), and *ACVR2A* (ID: Cf02664427_m1).

### 4.6. Gene Ontology, Pathway Analysis and Network Construction

Gene Ontology and pathway analysis of miRNA target genes were done using the Database for Annotation, Visualization and Integrated Discovery (DAVID) [48]. A common miRNA–TF interaction network was constructed between human and dog by analyzing the differentially expressed TFs from GSE31909. Briefly, we used TargetScan 7.2 [49] and miRDB [50] to predict miRNA targets. The common target genes between the two predictions were considered as targets for the respective miRNAs. A low FDR was considered to indicate a strong relationship between the annotation and the gene.

To construct a common miRNA–mRNA interaction network between humans and dogs we analyzed the BioProject GSE31909 datasets using the GEO2R tool (https://www.ncbi.nlm.nih.gov/geo/info/geo2r.html#background) to get the differentially expressed genes in human melanoma. We picked the target genes of the differentially expressed miRNAs from the differentially expressed genes in GSE31909. From the differentially expressed target genes, we only considered the TFs for network construction. We also considered the miRNAs that had the same seed sequences as the orthologous human miRNA sequences. The MSigDb gene families (http://software.broadinstitute.org/gsea/msigdb/index.jsp) were used to select the transcription factors (TFs) from the miRNA target genes. Since the expression of miRNAs and their targets are inversed, we built miRNA–TF co-regulatory networks with the inversely expressed TFs using Cytoscape v3.5 (http://www.cytoscape.org/) [51]. That means we built two networks, one with down-regulated miRNAs and up-regulated TFs and the other with up-regulated miRNAs and down-regulated TFs. Since TFs can regulate each other, we also performed a STRING v.10.5 (confidence score 0.700) (http://string-db.org/) [52] network analysis within each group of TFs. The final miRNA–TF regulatory network was obtained after merging the STRING TFs network with the respective miRNA–TF regulatory network in Cytoscape. The degree and betweenness of the network were measured using CentiScaPe 2.2 [53].

### 4.7. Statistical Analysis

We used GraphPad Prism 7 (www.graphpad.com) for the statistical analysis. Hierarchical clustering analysis was performed using the log_10_ value that was converted from the expression value of every miRNA from each sample. Unsupervised hierarchical clustering was done with Euclidean distance metric and complete linkage. Comparison of the qPCR data was done using Mann-Whitney test followed by Tukey’s test where appropriate. *p* values < 0.05 were considered significant. Correlation analysis was performed using Spearman’s correlation coefficient. For the miRNA chromosomal enrichment analysis, we used hyper-geometric test.

## 5. Conclusions

To the best of our knowledge, this study is the first report of comprehensively studied global miRNA profile about COM. Important miRNAs with respect to human melanoma was also explored in this study. As dogs are considered as models for human melanoma, further study will better explain the pathogenesis of melanoma in both species. Also key therapeutic option may reveal by the in-depth future study.

## Figures and Tables

**Figure 1 ijms-20-04832-f001:**
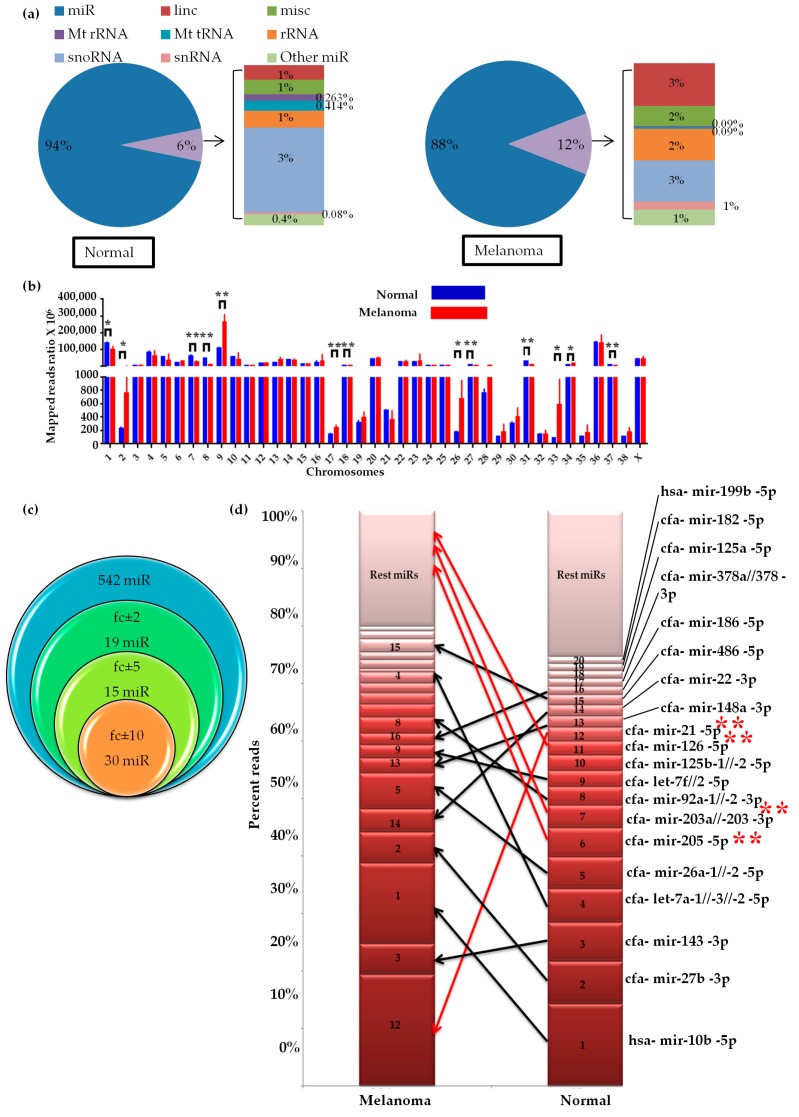
Profile of small RNA reads in canine oral melanoma by next-generation sequencing: (**a**) Percentages of the clean reads annotated under the different small RNA categories. Normal (*n* = 3), Melanoma (*n* = 8), (**b**) The miRNA reads were mapped to the canine genome (Canfam3.1) and the mapped reads ratio distributed in each chromosome was analyzed. Multiple t-test (many t-tests at once-one per row), * *p* < 0.05, ** *p* < 0.01, (**c**) Venn diagram showing the total numbers of identified and differentially expressed miRNAs in melanoma. First Venn (blue) indicates the total number of miRNAs identified in both groups. Other three Venn indicates the number of differentially expressed miRNAs (up (+) and down-regulated (−)). fc, fold change, (**d**) Top 20 highly expressed miRNAs in the normal and melanoma. Twelve miRNAs were common between the groups and rests eight were left blank in melanoma. In the bar graph, the number in each cell represents the rank of the miRNA with respect to the normal group. Arrows represent the changed position of miRNAs in melanoma group, Red arrows indicate the positions of significant deferentially expressed miRs. miRNA/s (miR/s), * statistically significant differentially expressed miRs between normal and melanoma.

**Figure 2 ijms-20-04832-f002:**
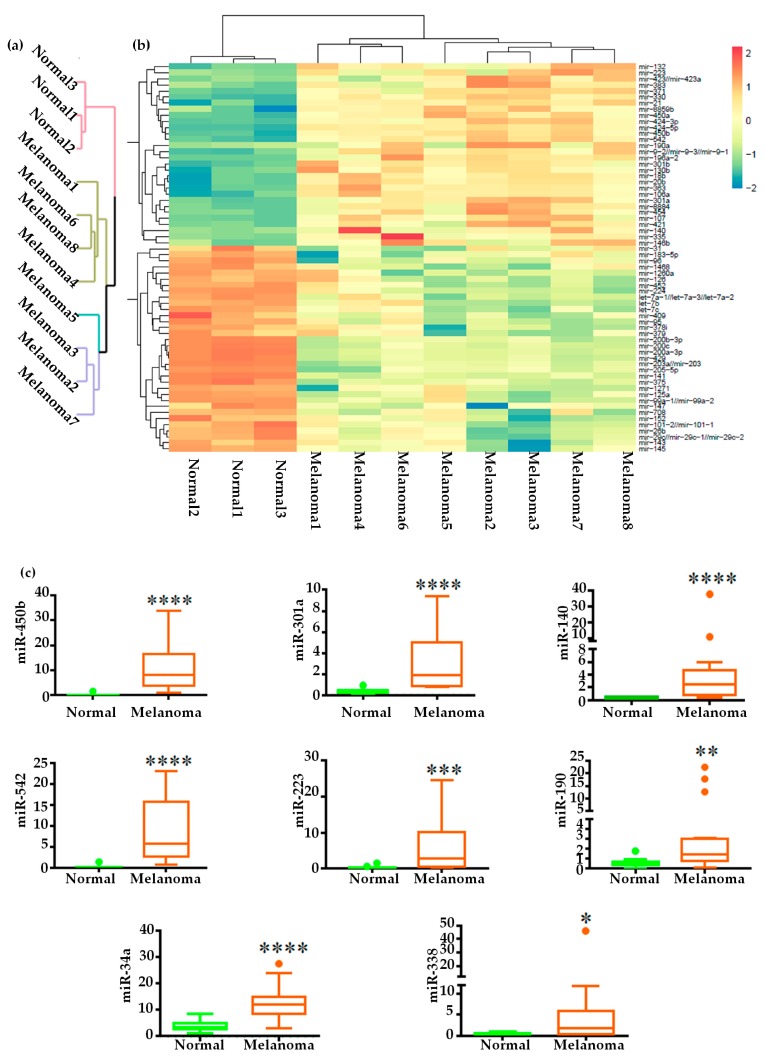
Differential miRNA expression in COM: (**a**) Unsupervised euclidean hierarchical clustering by the miRNA normalized expression values in the normal and melanoma libraries, (**b**) Heatmap visualizes the expression of statistically significant differentially expressed miRNAs in the normal and melanoma libraries. The colour scale (upper right) indicates the expression values. Up- and down-regulated miRNAs are shown in red to green, respectively. Colour saturation indicates the deviation from the median, (**c**) Relative expression of oncogenic miRNAs and (**d**) tumour suppressor miRNAs selected from the next-generation sequencing confirmed by qPCR. The Y-axes indicates the relative miRNA expression levels normalized against RNU6B (normal *n* = 12, melanoma *n* = 17, Mann-Whitney test followed by Tukey’s test, * *p* < 0.05, ** *p* < 0.01, *** *p* < 0.001, **** *p* < 0.0001), (**e**) Correlation of fold change between next-generation sequencing (NGS) and qPCR of the up- and down-regulated miRNAs. Each black circle in the graph represents a miRNA.

**Figure 3 ijms-20-04832-f003:**
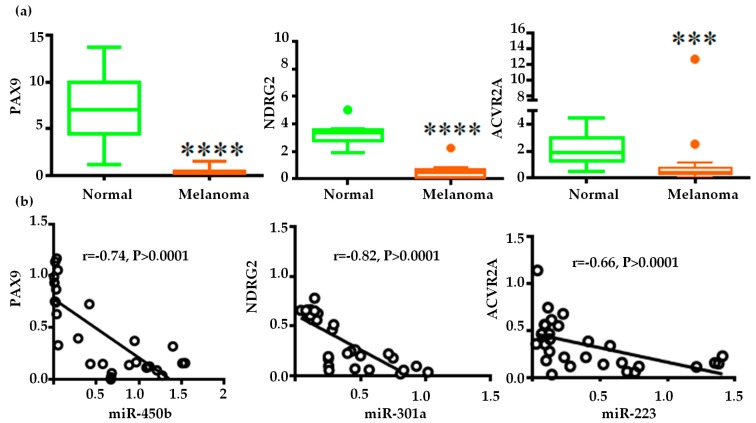
Gene regulatory function of miR-450b, miR-301a, and miR-223: (**a**) Relative expression of the target genes *PAX9*, *NDRG2*, and *ACVR2A*. Y-axes indicates the relative mRNA expression normalized against *GAPDH*, (**b**) Spearman correlation of the expression of the miR-450b-*PAX9*, miR-301a-*NDRG2*, and miR-223-*ACVR2A* pairs, (**c**) Relative expression of *BMP4* and *MMP9* in melanoma tissue samples, (**d**) Relative expression of miR-450b, *PAX9*, *BMP4*, and *MMP9* in the canine melanoma cell lines KMEC and LMEC. Mann-Whitney test followed by Tukey’s test, * *p* < 0.05, ** *p* < 0.01, *** *p* < 0.001, **** *p* < 0.0001. (**e**) Schematic representation of the miR-450b regulatory function. MiR-450b inhibits *PAX9* and, as a result, BMP4-MMP9 regulation is disrupted (↑–up-regulation, ⊺- Inhbition).

**Figure 4 ijms-20-04832-f004:**
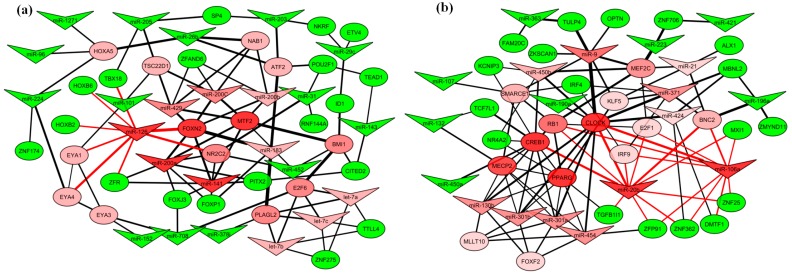
Common miRNA–transcription factor (TF) regulatory network between human and dog: (**a**) Regulatory network for down-regulated miRNAs and its up-regulated target TFs. MiR-126 influences eight TFs and has the highest centrality, (**b**) Regulatory network for up-regulated miRNAs and its down-regulated target TFs. MiR-20b and miR-106a have highest centralities. A miRNA (V-shaped) or TF (oval-shaped) is considered a node and line between nodes considered edge. Green and red indicate degree scores less and above than average and saturation shows deviation. Colour gradient of the nodes (miRNAs/mRNAs) indicates the strength of network regulation. Edge width represents edge betweenness. Node’s Edges with highest degree scores are in red.

**Figure 5 ijms-20-04832-f005:**
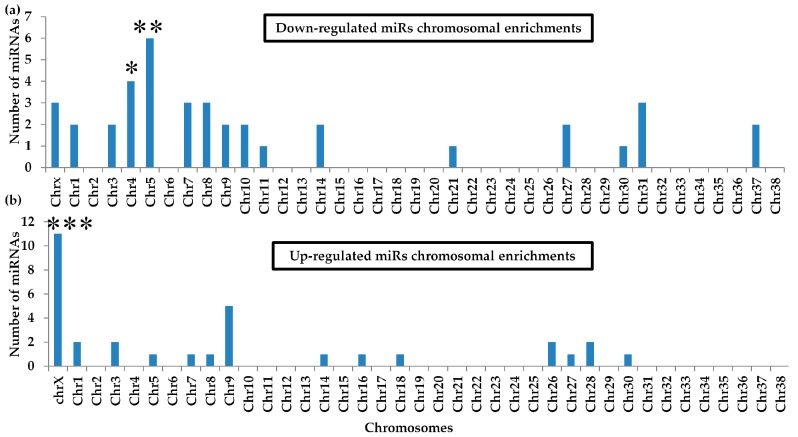
Chromosomal enrichment of differentially expressed miRNAs (miRs): (**a**) Chromosome enrichment of the down-regulated miRNAs, (**b**) Chromosome enrichment of the up-regulated miRNAs. Hypergeometric test, * *p* < 0.05, ** *p* < 0.01, *** *p* < 0.001.

**Table 1 ijms-20-04832-t001:** Gene ontology (GO) functional analysis of the target genes of differentially expressed miRNAs.

Down-Regulated miRNA’s Target Genes Ontology		
**Biological Process**		
**Terms**	FE ^1^	FDR ^2^
GO:0018105~peptidyl-serine phosphorylation	2.323	0.001
GO:0045944~positive regulation of transcription from RNA polymerase II promoter	1.461	0.025
**Cellular Component**		
GO:0005634~nucleus	1.277	1.65 × 10^−6^
GO:0005654~nucleoplasm	1.374	2.49 × 10^−5^
GO:0005737~cytoplasm	1.232	1.09 × 10^−4^
GO:0005911~cell-cell junction	2.11	0.068146
**Molecular Function**		
GO:0004702~receptor signaling protein serine/threonine kinase activity	2.932	9.98 × 10^−4^
GO:0005201~extracellular matrix structural constituent	3.373	0.002697
GO:0003682~chromatin binding	1.709	0.002892
GO:0003714~transcription corepressor activity	2.104	0.057312
**Up-Regulated miRNA’s Target Genes Ontology**		
**Biological Process**		
**Terms**	FE ^1^	FDR ^2^
GO:0042787~protein ubiquitination involved in ubiquitin-dependent protein catabolic process	2.208	0.004
**Cellular Component**		
GO:0005654~nucleoplasm	1.441	1.35 × 10^−7^
GO:0005737~cytoplasm	1.228	2.87 × 10^−4^
GO:0005794~Golgi apparatus	1.590	8.03 × 10^−4^
GO:0005634~nucleus	1.228	0.001041
**Molecular Function**		
GO:0061630~ubiquitin protein ligase activity	2.027	0.012
GO:0044212~transcription regulatory region DNA binding	2.127	0.019

^1^ Fold enrichment, ^2^ False discovery rate.

**Table 2 ijms-20-04832-t002:** KEGG pathway analysis of the target genes of the differentially expressed miRNAs.

Down-Regulated miRNAs Target Genes Pathway		
**Terms**	**FE ^1^**	**FDR ^2^**
cfa05206:MicroRNAs in cancer	2.522	6.72 × 10^−7^
cfa04010:MAPK signaling pathway	1.917	1.20 × 10^−4^
cfa04151:PI3K-Akt signaling pathway	1.762	1.65 × 10^−4^
cfa04360:Axon guidance	2.307	4.96 × 10^−4^
cfa05205:Proteoglycans in cancer	1.981	8.64 × 10^−4^
cfa04910:Insulin signaling pathway	2.209	8.88 × 10^−4^
cfa04152:AMPK signaling pathway	2.230	0.003
cfa04510:Focal adhesion	1.901	0.003
cfa04722:Neurotrophin signaling pathway	2.164	0.010
cfa04012:ErbB signaling pathway	2.384	0.013
cfa04512:ECM-receptor interaction	2.384	0.013
**Up-regulated miRNAs target genes pathway**		
**Terms**	FE ^1^	FDR ^2^
cfa04144:Endocytosis	2.424	1.31 × 10^−11^
cfa04810:Regulation of actin cytoskeleton	2.273	2.78 × 10^−7^
cfa05200:Pathways in cancer	1.604	0.010
cfa04710:Circadian rhythm	3.536	0.060
cfa05410:Hypertrophic cardiomyopathy (HCM)	2.398	0.063
cfa05414:Dilated cardiomyopathy	2.346	0.065
cfa04713:Circadian entrainment	2.210	0.098

^1^ Fold enrichment. ^2^ False discovery rate.

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
