# Peer review of "Micro RNA Transcriptome Profile in Canine Oral Melanoma"

_ijms, 2019, doi:10.3390/ijms20194832_

Round 1

Reviewer 1 Report

In this paper, the authors analyze the expression of microRNAs in canine oral melanoma. They identify miRNA expressed in tumors and link them to target genes and pathways. While of potential importance, the analysis is rather basic, the presentation is lacking and often unclear, and the significance of the results not fully established.

Figure 1b: the analysis of number of reads per chromosome provides little information. For example, a single miRNA can explain the entire difference in one chromosome. Alternatively, total number of reads on a chromosome could be the same while the expression of different miRNAs on that chromosome may vary between the samples. A miRNA-specific analysis should replace this.

Figure 1c and text- unclear- is this the number of miRNA in “both groups” (text) or those that are differentially expressed (figure legend) (and if so, are they all induced or are some repressed)?

Figure 1d- this is not the best way to represent this information. Ranking is a semi-quantitative (in other words, gross) analysis. Better would be to compare the expression levels of each miR in the two samples. Text mentions differences in ranking, however it should be established which of these are statistically significant. 

Figure 2: a and b should be combined; the correlation graph should be arranged based on the clustering and the corresponding dendrogram shown. Genes should also be clustered, not just samples. Melanoma samples appear to form two clusters in figure 2a (samples 1,6,8,4 and samples 3,2,7 (and 5)) but this is not apparent in the correlation matrix. A clustered correlation matrix with all genes should also be shown; showing only the differentially expressed genes will by definition cluster normal samples separately from melanoma samples. 

Section 2.5: “Target genes of the down-regulated miRNAs pretend to be oncogenic”. The remainder of this section also seems to be pretend: listing general GO categories is hardly evidence of them being related to cancer. Any category could be claimed to be linked to cancer. Instead, the authors should compare their gene list to the Cancer Gene Census or similar lists.

Section 2.6 (and to some extend the manuscript in general) is descriptive, with no clear take home message.

Section 2.7- also seems descriptive. Why does it matter on which chromosomes miR reside? It’s also unclear how this analysis relates to figure 1b- difficult to reconcile the two.

Last sentence of discussion “Besides, our findings give an insight the basic fundamentals of TWT and mucosal melanoma” is poorly written, reflecting the general feeling throughout this manuscript that this is still an incomplete presentation. 

Minor comments:

Introduction: “microRNAs (miRNAs) are now in phase I trials to treat human” - this sentence should be detailed (e.g. to treat human melanoma?) and referenced.

Figure 1b: figure legend says “average percentage of reads” but y axis scale is number (although the label is “ratio”)- please clarify. Also please clarify “multiple” t test in the figure legend.

Section 2.4: clarify “We selected three miRNAs (miR-450b, miR-301a and miR-223) to know their gene regulatory function in reference with the human study”. What human study?

Reviewer 2 Report

In the present manuscript entitled “Micro RNA transcriptome profile in canine oral melanoma”, Rhaman and collaborators provide a study of the differences between melanoma and normal mucosae from dogs. The description of such differences might be of interest to determine whether dogs are suitable models to study the disease itself or potential therapies in the context of oral melanoma. Even though the research is well conducted and there are some potentially relevant results, several aspects need certain improvement.

The authors must proof read the entire manuscript as one can observe several typos through it. Take as examples line 425 “Cytoscope v3.5” instead of Cytoscape v3.5, and the extra a in “miRaBASE” (Suppl. Figure 1b). Moreover, the document will benefit from a grammatical revision in order to improve its readability. The main objective/-s of the study should be clearly stated in the introduction. In the introduction the authors stated that microRNAs are in phase I studies to treat humans. Please provide meaningful references and examples supporting that statement. There is no information about the control animals used in the study; supplementary table 1 shows only the disease animals. The authors recognize the limitation of having only three samples as controls for the NGS. However, a different background of the animals can exacerbate this limitation, and the authors need to discuss this aspect. The authors must improve the results section. They narrow down their number of microRNAs to three relevant ones (even providing a hypothetical pathway involving cfa-miR-450b, PAX9 and MMP9), and then they come back to the complete set of differentially expressed genes. That can be confusing for the reader and should be improved. Please, provide a reasonable reasoning behind the selection of the validated microRNAs. In addition, figure 2 is too busy with panels for the different microRNAs. The authors should choose which microRNAs they want to show (placing the rest in the supplementary material) or look for an alternative to improve visualization. Did the authors find any microRNA that has not been validated? If that is the case, it will be informative to show it. Figure 2e shows a correlation between NGS and PCR but it is unclear which results are shown in the image. Are these data from a single microRNA or a combination of them? In the same way, the selection of the target genes is unclear. Please, explain further why those genes were selected. What do the authors mean with "downstream regulator" when referring to BMP4 in line 152? It is difficult to understand that the microRNAs showing the highest degree of centrality are not analysed during the article. Indeed, cfa-miR-106a has not been validated by PCR. Please, justify. Related to figure 3, the authors must be careful with their conclusions as they do not show strong evidences supporting their conclusion about the pathway involving cfa-miR-450b The annotation of table 1 (referred in the text as table 1a) is confusing. Please, consider to split the table in two. Moreover, the authors stated that the upregulated genes were related to the ubiquitination (line 185-186). However, the reader cannot extract this information from the actual table. The chapter 2.7 of the results section seems disconnected from the rest of the article. Please, justify the reasoning to study the chromosomal enrichment at the beginning of the chapter or in the introduction. Supplementary table 2 is difficult to interpret, because the authors did not sort the data.

Author Response

Response to Reviewer 2 Comments

In the present manuscript entitled “Micro RNA transcriptome profile in canine oral melanoma”, Rhaman and collaborators provide a study of the differences between melanoma and normal mucosae from dogs. The description of such differences might be of interest to determine whether dogs are suitable models to study the disease itself or potential therapies in the context of oral melanoma. Even though the research is well conducted and there are some potentially relevant results, several aspects need certain improvement.

Author: We thank the reviewer for the nice enthusiastic words. We appreciate your suggestion and comments. We tried to answer and follow your question and comments accordingly.

The authors must proof read the entire manuscript as one can observe several typos through it. Take as examples line 425 “Cytoscope v3.5” instead of Cytoscape v3.5, and the extra a in “miRaBASE” (Suppl. Figure 1b).

Author: We made the correction and proof read.

Moreover, the document will benefit from a grammatical revision in order to improve its readability.

Author: We did the English editing with native English speaker but if we need again, we are happy to do it.

The main objective/-s of the study should be clearly stated in the introduction.

Author: We appreciate for your clear and helpful suggestion. We added the objectives in the introduction. You can find in Line 60-63

In the introduction the authors stated that microRNAs are in phase I studies to treat humans. Please provide meaningful references and examples supporting that statement.

Author: We provided the reference and mentioned the disease name. Please find the change in line 52-54

There is no information about the control animals used in the study; supplementary table 1 shows only the disease animals.

Author: We added the information of control animal in the text. Line 377-379

The authors recognize the limitation of having only three samples as controls for the NGS. However, a different background of the animals can exacerbate this limitation, and the authors need to discuss this aspect.

Author: All the normal oral tissues were obtained from healthy laboratory beagle dogs which minimize the limitation of using normal samples from diverse background animals. We added in the discussion in line 365-366

The authors must improve the results section. They narrow down their number of microRNAs to three relevant ones (even providing a hypothetical pathway involving cfa-miR-450b, PAX9 and MMP9), and then they come back to the complete set of differentially expressed genes. That can be confusing for the reader and should be improved.

Author: We wish to thanks reviewer for the suggestion. We have tried to improve the section. We are very happy to accept any further suggestions. You can find the change in Line 161-169, 170-173, 178-181, 188-191

 Please, provide a reasonable reasoning behind the selection of the validated microRNAs. In addition, figure 2 is too busy with panels for the different microRNAs. The authors should choose which microRNAs they want to show (placing the rest in the supplementary material) or look for an alternative to improve visualization. Did the authors find any microRNA that has not been validated? If that is the case, it will be informative to show it.

Author: We appreciate your kind and helpful suggestion. We selected the miRNAs on the basis of three different criteria for validation.

Significantly differentially expressed miRNAs those were not reported or validated previously (miR-450b, 301a, 140, 542, 223, 190, 429, 96, 375,183, 200b, 141). Significantly differentially expressed miRNAs those were reported or validated previously (miR-21, miR-122, 383 and 143) (1). miRNAs those were beyond the stringent filtering criteria that means they were not able to fulfill (Fold change>2, FDR <0.05 and miRNA mean read counts in either normal or melanoma >50) all the three criteria or very close numerically to the filtering criteria  (miR-122, miR-34a, miR-338, miR-182, and miR-1). 

According to the reviewer suggestion we placed previously reported miRNAs in the supplementary figure 2.  Also we have tried to improve the visualization of the figures.

 Figure 2e shows a correlation between NGS and PCR but it is unclear which results are shown in the image. Are these data from a single microRNA or a combination of them?

Author: Each black circle in the graph re-presents a miRNA. We calculated the fold change of PCR and NGS of each miRNAs and counted the co-relation of the fold changes. We rearranged the graph for more visibility and made correction on the legend.

In the same way, the selection of the target genes is unclear. Please, explain further why those genes were selected.

Author: To know the differentially expressed miRNAs function we need to focus on the expression of their respective target genes. However, it is unfeasible to explore all the miRNAs targets in a single study. So, we wanted to investigate the function of the miR-450b, 301a and 223 in canine oral melanoma (COM). We chose these three miRNA because to our knowledge miR-450b had no report in melanoma and other two miRNAs were less studied in human melanoma and no report in COM. Moreover, PAX9 and NDRG2 reported only in human and mouse melanoma respectively and there was no report about ACVR2A in melanoma (2-3). On the other hand, binding site of the miR-450b-PAX9, miR-301a-NDRG2 and miR-223-ACVR2A are conserved between human and dog (Figure S3). Binding of the respective miRNA-mRNA pairs were confirmed experimentally in the individual human study (4-6). So, we chose these three genes because we hypothesized that similar phenomenon (miRNA-mRNA binding) may exist in the canine oral melanoma. As in our experiment miR-450b, 301a and 223 were up-regulated so if our hypothesis is true the target genes must be down-regulated and they should have a strong negative co-relation with the respective pairs. We found as we expected which were reported in the study. So our study reveals, the expression of these three genes (PAX9, NDRG2 and ACVR2A) is down-regulated in COM. Suppression of the PAX9, NDRG2 and ACVR2A may be occurred due to up-regulation of the miR-450b, 301a and 223 respectively. Moreover, previous human study reported that PAX9 was down-regulated and miR-301a, 223 were up-regulated in human melanoma (2, 7-8). So it also indicates that there are similarities between human and canine melanoma in respect of PAX9, miR-301a and 223 expressions.  

What do the authors mean with "downstream regulator" when referring to BMP4 in line 152?

Author: Previous study reported that BMP4 act as a downstream of PAX9. In our text with downstream regulator we mean PAX9 can regulate/control the BMP4 expression. However, we replaced the “downstream regulator” with “downstream” because the word “regulator” may create confusion. Line 327

It is difficult to understand that the microRNAs showing the highest degree of centrality are not analysed during the article. Indeed, cfa-miR-106a has not been validated by PCR. Please, justify.

Author: Down-regulation of miR-126 was reported previously in COM (9). Moreover, miR-20b and miR-106a up-regulation was also reported in human melanoma (10, 11). However, it might also be interesting to investigate the comparative study of transcription factor regulation of these miRNAs between human and dog. We will address this issue in our future study.

Related to figure 3, the authors must be careful with their conclusions as they do not show strong evidences supporting their conclusion about the pathway involving cfa-miR-450b.

Author: We appreciate your kind suggestion. We modified the conclusion. Please find the correction in Line 188-191, 329-331

 The annotation of table 1 (referred in the text as table 1a) is confusing. Please, consider to split the table in two. Moreover, the authors stated that the upregulated genes were related to the ubiquitination (line 185-186). However, the reader cannot extract this information from the actual table.

Author: We made the correction and split the table according to down and up-regulated miRNA’s target gene ontology. We also changed the text regarding to the up-regulated miRNAs target genes ontology related function. Line 213-218

 The chapter 2.7 of the results section seems disconnected from the rest of the article. Please, justify the reasoning to study the chromosomal enrichment at the beginning of the chapter or in the introduction.

Author: We appreciate your kind and helpful suggestion. We added the justification in the beginning of the chapter (chapter 2.7). Line 267-272

 Supplementary table 2 is difficult to interpret, because the authors did not sort the data.

Author: We shorted the fold change and FDR.

 References

Ushio, N., Rahman, M.M., Maemura, T., Lai, Y.C., Iwanaga, T., Kawaguchi, H., Miyoshi, N., Momoi, Y. and Miura, N., 2019. Identification of dysregulated microRNAs in canine malignant melanoma. Oncology letters17(1), pp.1080-1088. Hata, S., Hamada, J.I., Maeda, K., Murai, T., Tada, M., Furukawa, H., Tsutsumida, A., Saito, A., Yamamoto, Y. and Moriuchi, T., 2008. PAX4 has the potential to function as a tumor suppressor in human melanoma. International journal of oncology33(5), pp.1065-1071. Kim, A., Yang, Y., Lee, M.S., Yoo, Y.D., Lee, H.G. and Lim, J.S., 2008. NDRG2 gene expression in B16F10 melanoma cells restrains melanogenesis via inhibition of Mitf expression. Pigment cell & melanoma research21(6), pp.653-664. Sun, M.M., Li, J.F., Guo, L.L., Xiao, H.T., Dong, L., Wang, F., Huang, F.B., Cao, D., Qin, T., Yin, X.H. and Li, J.M., 2014. TGF-β1 suppression of microRNA-450b-5p expression: a novel mechanism for blocking myogenic differentiation of rhabdomyosarcoma. Oncogene33(16), p.2075. Guo, Y.J., Liu, J.X. and Guan, Y.W., 2016. Hypoxia induced upregulation of miR-301a/b contributes to increased cell autophagy and viability of prostate cancer cells by targeting NDRG2. Eur Rev Med Pharmacol Sci20(1), pp.101-8. Yang, L., Li, Y., Wang, X., Mu, X., Qin, D., Huang, W., Alshahrani, S., Nieman, M., Peng, J., Essandoh, K. and Peng, T., 2016. Overexpression of miR-223 tips the balance of pro-and anti-hypertrophic signaling cascades toward physiologic cardiac hypertrophy. Journal of Biological Chemistry291(30), pp.15700-15713. Cui, L., Li, Y., Lv, X., Li, J., Wang, X., Lei, Z. and Li, X., 2016. Expression of MicroRNA-301a and its functional roles in malignant melanoma. Cellular Physiology and Biochemistry40(1-2), pp.230-244. Sand, M., Skrygan, M., Sand, D., Georgas, D., Gambichler, T., Hahn, S.A., Altmeyer, P. and Bechara, F.G., 2013. Comparative microarray analysis of microRNA expression profiles in primary cutaneous malignant melanoma, cutaneous malignant melanoma metastases, and benign melanocytic nevi. Cell and tissue research351(1), pp.85-98. Noguchi, S., Mori, T., Hoshino, Y., Yamada, N., Maruo, K. and Akao, Y., 2013. MicroRNAs as tumour suppressors in canine and human melanoma cells and as a prognostic factor in canine melanomas. Veterinary and comparative oncology11(2), pp.113-123. Xu, Y., Brenn, T., Brown, E.R.S., Doherty, V. and Melton, D.W., 2012. Differential expression of microRNAs during melanoma progression: miR-200c, miR-205 and miR-211 are downregulated in melanoma and act as tumour suppressors. British journal of cancer106(3), p.553. Palkina, N.V., Komina, A.V., Aksenenko, M.B. and Ruksha, T.G., 2017. The pro-oncogenic effect of miR-106a microRNA inhibition in melanoma cells in vitro. Cell and Tissue Biology11(1), pp.1-8.

 Reviewer 3 Report

Mahfuzur Rahman and co-workers present a research article describing differential miRNA profiling in canine oral melanoma (COM). Overall the article is well written and should provide a valuable resource for the field of molecular oncology using dog as a model. This study is hardly bioinformatic bias, however it also shows some qRT-PCR that nicely confirmed the NGS data and the downregulation of the miRNA targets. The authors report 30 differential oncogenic miRNA present in COM samples compared against healthy tissue samples. They also found three oncogenic miRNA targets that were down-regulated in COM.  They focus on miR-450b and propose a negative regulatory function of PAX9 and BMP4 over MMP9. At the end, I could not find any connection with their story and the jump into the chromosomal enrichment, here they came back to look for the position all their differential miRNA on the genome. They found X chromosome was enriched with oncogenic miRNAs. Interestingly as they mentioned in the discussion, females have a higher incidence in mucosal melanoma and breast cancer X-linked miRNAs but how this will impact their own research, they have both male and female samples, but no analysis were performed to try to see if differences in miRNAs reflect on sex. Minor points must be addressed before going any further in the publication process:

In line 77-798, authors claim the percentage of other ncRNA was twice more in COM and refer to Fig.1a however in Fig1a the pie charts and barplots showing the profile of small RNA reads do not have that category (maybe they refer to Other miR?).

Line 86 talks about how they get their top 20 miRs and Fig 1d present this list but no scores are shown. Would be worth to add the score value that they get together with the mentioned position. Also, why this table does not correlate with the positions on the heatmap Fig 2b?

Line 119 shows also the selection of 20 differentially expressed miRNAs as candidates for the qPCR validation, please specify based on which score this selection was base on?

Regarding the validation by RT-qPCR of miRNAs, please specify which products are you targeting: pri-mir, pre-mir, mature miRNA, miR loaded (active)? This information is of great value for the readers due to the complex biogenesis and mechanisms of function of miRNAs. And will help to support (or not?) their conclusions. 

Further in the manuscript in the section of “Gene regulatory function of oncogenic miRNAs”, I agree that their RT-qPCR really showed a significant negative correlation between the three selected miRNAs (miR-450b, miR-301a, and miR-223) and their targets. However, correlation does not imply causation, maybe for future studies reverse genetic experiments (overexpression of miRNAs, KO/silencing of miRNAs, western blots of the proteins targeted?) need to be performed to probe this association.

In fig. 4 line 219. The figure is confusing and does not provide any valuable additional information. 

In line 237 and Fig 5. Why x-axes are not the same? Is difficult to say something about the data if data is missing. All chromosomes should be shown. 

Once these minor issues addressed, it will my pleasure to support the acceptance of this article for publication in  Int. J. Mol. Sci.

Author Response

Response to Reviewer 3 Comments

Mahfuzur Rahman and co-workers present a research article describing differential miRNA profiling in canine oral melanoma (COM). Overall the article is well written and should provide a valuable resource for the field of molecular oncology using dog as a model. This study is hardly bioinformatic bias, however it also shows some qRT-PCR that nicely confirmed the NGS data and the downregulation of the miRNA targets. The authors report 30 differential oncogenic miRNA present in COM samples compared against healthy tissue samples. They also found three oncogenic miRNA targets that were down-regulated in COM.  They focus on miR-450b and propose a negative regulatory function of PAX9 and BMP4 over MMP9.

Author: Thank you and we appreciate the positive feedback from the reviewer. We tried to answer and follow your question and comments accordingly.

At the end, I could not find any connection with their story and the jump into the chromosomal enrichment, here they came back to look for the position all their differential miRNA on the genome.

Author: We added the clarification to analyze the chromosomal enrichment of the differentially expressed miRNAs in the beginning of the chapter 2.7. Also the section was rearranged. Line 267-272, 282-283, 292-299  

They found X chromosome was enriched with oncogenic miRNAs. Interestingly as they mentioned in the discussion, females have a higher incidence in mucosal melanoma and breast cancer X-linked miRNAs but how this will impact their own research, they have both male and female samples, but no analysis were performed to try to see if differences in miRNAs reflect on sex.

Author: It might be interesting to study sex difference in canine oral melanoma. However, sex differences studies related to diseases consisted of a large sample size which is not present in our study (1-3). Moreover, exact sex difference study in dog may appear difficult because unlike human, dog are often neutered or spayed in the early age. 

Minor points must be addressed before going any further in the publication process:

In line 77-798, authors claim the percentage of other ncRNA was twice more in COM and refer to Fig.1a however in Fig1a the pie charts and barplots showing the profile of small RNA reads do not have that category (maybe they refer to Other miR?).

Author: We are sorry for the confusion. We referred to the percentage other ncRNA (Mt-rRNA, Mt-tRNA, linc-RNA, sno-RNA, snRNA, misc-RNA, rRNA, other miR) except miRNA.  For example, In case of melanoma, miRs=88%, other=12% (Mt-rRNA, Mt-tRNA, linc-RNA, sno-RNA, snRNA, misc-RNA, rRNA, other miR). We made the correction in the text. Line 80-83

Line 86 talks about how they get their top 20 miRs and Fig 1d present this list but no scores are shown. Would be worth to add the score value that they get together with the mentioned position. Also, why this table does not correlate with the positions on the heatmap Fig 2b?

Author: We appreciate your comment. We modified this figure to increase visibility. Moreover, we considered the miRNA’s expression to give the rank (similar to score). MiRNA which have the highest expression ranked 1, others were ranked consecutively on the basis of their expression. We have changed the text. Line 91

We created the heatmap with our differentially expressed significant (Fold change>2, FDR <0.05 and miRNA mean read counts in either normal or melanoma >50) miRNAs list. However, table in the previous figure 1d represented the top 20 highly expressed miRNAs in the normal group. In the new figure 1d we marked the significant differentially expressed miRNAs with star (**) sign.

Line 119 shows also the selection of 20 differentially expressed miRNAs as candidates for the qPCR validation, please specify based on which score this selection was base on?

Author: We selected the miRNAs on the basis of three different criteria for validation.

Significantly differentially expressed miRNAs those were not reported or validated previously (miR-450b, 301a, 140, 542, 223, 190, 429, 96, 375,183, 200b, 141). Significantly differentially expressed miRNAs those were reported or validated previously (miR-21, miR-122, 383 and 143) (4). miRNAs those were beyond the stringent filtering criteria that means they were not able to fulfill (Fold change>2, FDR <0.05 and miRNA mean read counts in either normal or melanoma >50) all the three criteria or very close numerically to the filtering criteria (miR-122, miR-34a, miR-338, miR-182, and miR-1). 

Regarding the validation by RT-qPCR of miRNAs, please specify which products are you targeting: pri-mir, pre-mir, mature miRNA, miR loaded (active)? This information is of great value for the readers due to the complex biogenesis and mechanisms of function of miRNAs. And will help to support (or not?) their conclusions. 

Author: The TaqMan microRNA primers used in our study can quantitate only mature miRNAs. We added the information in the Materials and Methods in line 451-452

Further in the manuscript in the section of “Gene regulatory function of oncogenic miRNAs”, I agree that their RT-qPCR really showed a significant negative correlation between the three selected miRNAs (miR-450b, miR-301a, and miR-223) and their targets. However, correlation does not imply causation, maybe for future studies reverse genetic experiments (overexpression of miRNAs, KO/silencing of miRNAs, western blots of the proteins targeted?) need to be performed to probe this association.

Author: We appreciate the reviewer comments. Although, miRNA and their target mRNA binding were experimentally validated previously (5-7). We agree with the reviewer comments that we will consider reverse genetics experiments in our future study to explain the association of these miRNAs and their targets more clearly.

In fig. 4 line 219. The figure is confusing and does not provide any valuable additional information. 

Author: In this figure we tried to represent the miRNA that target transcription factor (TFs) in each group. Through this figure we can easily observe that miR-126, miR-20b, and miR-106a target highest number of TFs in their respective group (connected with red line). Also, we can see the target TFs and miRNAs names and interconnection in the same figure at a glance. Please appreciate our effort.    

In line 237 and Fig 5. Why x-axes are not the same? Is difficult to say something about the data if data is missing. All chromosomes should be shown. 

Author: We added all the chromosome in the X-axis in our new figure.

Once these minor issues addressed, it will my pleasure to support the acceptance of this article for publication in  Int. J. Mol. Sci.

 References

Scoggins, C.R., Ross, M.I., Reintgen, D.S., Noyes, R.D., Goydos, J.S., Beitsch, P.D., Urist, M.M., Ariyan, S., Sussman, J.J., Edwards, M.J. and Chagpar, A.B., 2006. Gender-related differences in outcome for melanoma patients. Annals of surgery243(5), p.693. McLaughlin, C.C., Wu, X.C., Jemal, A., Martin, H.J., Roche, L.M. and Chen, V.W., 2005. Incidence of noncutaneous melanomas in the US. Cancer: Interdisciplinary International Journal of the American Cancer Society103(5), pp.1000-1007. Yang, W., Warrington, N.M., Taylor, S.J., Whitmire, P., Carrasco, E., Singleton, K.W., Wu, N., Lathia, J.D., Berens, M.E., Kim, A.H. and Barnholtz-Sloan, J.S., 2019. Sex differences in GBM revealed by analysis of patient imaging, transcriptome, and survival data. Science translational medicine11(473), p.eaao5253. Ushio, N., Rahman, M.M., Maemura, T., Lai, Y.C., Iwanaga, T., Kawaguchi, H., Miyoshi, N., Momoi, Y. and Miura, N., 2019. Identification of dysregulated microRNAs in canine malignant melanoma. Oncology letters17(1), pp.1080-1088. Sun, M.M., Li, J.F., Guo, L.L., Xiao, H.T., Dong, L., Wang, F., Huang, F.B., Cao, D., Qin, T., Yin, X.H. and Li, J.M., 2014. TGF-β1 suppression of microRNA-450b-5p expression: a novel mechanism for blocking myogenic differentiation of rhabdomyosarcoma. Oncogene33(16), p.2075. Guo, Y.J., Liu, J.X. and Guan, Y.W., 2016. Hypoxia induced upregulation of miR-301a/b contributes to increased cell autophagy and viability of prostate cancer cells by targeting NDRG2. Eur Rev Med Pharmacol Sci20(1), pp.101-8. Yang, L., Li, Y., Wang, X., Mu, X., Qin, D., Huang, W., Alshahrani, S., Nieman, M., Peng, J., Essandoh, K. and Peng, T., 2016. Overexpression of miR-223 tips the balance of pro-and anti-hypertrophic signaling cascades toward physiologic cardiac hypertrophy. Journal of Biological Chemistry291(30), pp.15700-15713.

Round 2

Reviewer 1 Report

The authors have addressed most of my concerns.

Reviewer 2 Report

In the present manuscript entitled “Micro RNA transcriptome profile in canine oral melanoma”, Rhaman and collaborators provide a study of the differences between melanoma and normal mucosae from dogs. The description of such differences might be of interest to determine whether dogs are suitable models to study the disease itself or potential therapies in the context of oral melanoma. Even though the research is rather well conducted and there are some potentially relevant results, several aspects need certain improvement.

The authors could still improve the general writing through the manuscript. What do the authors mean with “other miRs” in line 82? The main objective/-s of the study seem rather general. The manuscript will benefit from a bit more focus. The control animals used in the study presented the same background, which reduces the variability in the control group. However, the animals for the COM group are from different breeds and, if well understood from the table, none of them are Beagle dogs. In addition, there is no mention about the sex of the dogs in the control group, incorporating sex as a potential source of bias in the results. The authors recognize the limitation of having only three samples in the controls but they underestimate the different background of the animals as a big limitation of the study. The authors must improve the results section. The authors should narrow the results that they want to show for this specific publication. Some of the results are disconnected from the rest of the manuscript and they seem more like an introduction for a future project. The reasoning behind the selection of the validated microRNAs is vague. Even though the authors provide three selection criteria, they are not clearly separated. To illustrate that, notice that the two first criteria are fully complementary, so all the significant microRNAs should be included in the analysis. The authors need to improve figure 2 because it is too long. In figure 2e is unclear which results are shown in the image as they are less circles than microRNAs tested (or the circles are perfectly overlapped). In the same way, the selection of the target genes seems random. The authors stated “it is unfeasible to explore all the miRNAs targets in a single study”. However they are doing a similar analysis for figure 4. Moreover, the authors are doing a search for transcription factors relevant for their study but the main transcription factor that they study (PAX9) it does not appear in that analysis (at least not in figure 4). The authors must be cautious with their conclusions as they do not show strong evidences supporting a direct interaction between cfa-miR-450b and PAX9; or an effect of the microRNAs on the pathway involving PAX9, BMP4, and MMP9. More experiments will be needed to prove that the correlations shown are due to a direct interaction of the different elements.